# Bipolar Disorder and Immune Dysfunction: Epidemiological Findings, Proposed Pathophysiology and Clinical Implications

**DOI:** 10.3390/brainsci7110144

**Published:** 2017-10-30

**Authors:** Joshua D. Rosenblat, Roger S. McIntyre

**Affiliations:** 1Department of Psychiatry, University of Toronto, Toronto, ON M5T 2S8, Canada; 2Mood Disorders Psychopharmacology Unit, University Health Network, 399 Bathurst Street, MP 9-325, Toronto, ON M5T 2S8, Canada; Roger.McIntyre@uhn.ca

**Keywords:** bipolar disorder, inflammation, cytokines, depression, neuroprogression, cognition, n-acetylcysteine, infliximab, celecoxib, minocycline

## Abstract

Bipolar disorder (BD) is strongly associated with immune dysfunction. Replicated epidemiological studies have demonstrated that BD has high rates of inflammatory medical comorbidities, including autoimmune disorders, chronic infections, cardiovascular disease and metabolic disorders. Cytokine studies have demonstrated that BD is associated with chronic low-grade inflammation with further increases in pro-inflammatory cytokine levels during mood episodes. Several mechanisms have been identified to explain the bidirectional relationship between BD and immune dysfunction. Key mechanisms include cytokine-induced monoamine changes, increased oxidative stress, pathological microglial over-activation, hypothalamic-pituitary-adrenal (HPA) axis over-activation, alterations of the microbiome-gut-brain axis and sleep-related immune changes. The inflammatory-mood pathway presents several potential novel targets in the treatment of BD. Several proof-of-concept clinical trials have shown a positive effect of anti-inflammatory agents in the treatment of BD; however, further research is needed to determine the clinical utility of these treatments. Immune dysfunction is likely to only play a role in a *subset* of BD patients and as such, future clinical trials should also strive to identify which specific group(s) of BD patients may benefit from anti-inflammatory treatments.

## 1. Introduction

Bipolar disorder (BD) is a severe and persistent mental illness associated with significant morbidity and mortality. While numerous hypotheses have been proposed to explain the underlying patho-etiology of BD, the mechanisms sub-serving disease onset and progression remain largely unknown. More recently, immune dysfunction has been implicated in the patho-etiology of BD [1]. The hypothesis that immune dysfunction may be a mediator of disease progression in BD was first proposed by Horrobin & Lieb (1981) [2] who hypothesized that immune modulation may be a key mechanism of action in lithium’s mood stabilizing effects. They further hypothesized that the relapsing-remitting nature of BD may be driven by the immune system, as seen in other relapsing-remitting inflammatory disorders, such as multiple sclerosis (MS) [2]. Since their hypothesis was proposed, numerous investigators have studied the interaction between BD and immune dysfunction [1,3,4,5].

The primary aim of the current review is to summarize and synthesize studies assessing the interaction between BD and immune dysfunction. Towards this end, we will summarize the following key areas: (1) epidemiological data revealing high rates of comorbidity between BD and inflammatory disorders; (2) cytokine studies showing increased central and peripheral levels of pro-inflammatory molecules in BD compared to healthy controls; (3) proposed pathophysiological mechanisms sub-serving the inflammatory-mood pathway and (4) clinical implications of the interaction between BD and immune dysfunction, with a focus on repurposing anti-inflammatory agents in the treatment of bipolar depression.

Of note, the current review is not a systematic review, but rather narrative in nature, to provide a broad overview of the topic. A systematic review was not conducted given the breadth of the topic and vast number of studies on the various elements of the interactions between BD and inflammation. As such, the authors decided to focus on particularly relevant studies rather than exhaustively reviewing all published articles. The authors acknowledge that this approach is vulnerable to the presentation of a biased perspective; however, have attempted to present in an unbiased manner, highlighting areas of controversy and disagreement when needed.

## 2. Bipolar Disorder and Inflammatory Comorbidities

One potential indicator to suggest an interaction between BD and immune dysfunction is the high rates of inflammatory medical comorbidities in BD [6]. The association between BD and inflammatory comorbidities has been well established in numerous epidemiological studies; however, the direction of causality remains somewhat unclear. As shown in Figure 1a, immune dysfunction may be a common underlying cause of both BD and an inflammatory comorbidity in a given patient. Alternatively, BD may proceed the inflammatory condition or vice versa (Figure 1b,c). All three scenarios are observed in the BD population suggesting that the interaction is likely bidirectional in that immune dysfunction, BD and inflammatory comorbidities may be perpetuating each other as shown in Figure 1d [6]. Further, genetic and environmental risk factors for immune dysfunction may simultaneously increase the risk of developing both BD and other inflammatory comorbidities. Herein we summarize pertinent epidemiological findings showing the association between BD and inflammatory comorbidities.

When an inflammatory comorbidity is present, peripherally released pro-inflammatory cytokines may increase systemic cytokine levels (e.g., IL-2, IL-6, TNF-α) throughout the body, including in the brain [7]. The subtler effects of chronic low grade systemic inflammation on off-target areas (e.g., the brain) has been increasingly recognized as important [8,9]. Admittedly, the association between BD and inflammatory comorbidities does not, in itself, prove causation, however, the biological mechanisms [10] to be further discussed in Section 4 provide further evidence that these epidemiological observations (summarized in Table 1) are likely to be more than just spurious associations.

### 2.1. Bipolar Disorder and Autoimmune Disorders

Autoimmune disorders represent the most “classic” of inflammatory conditions in that they are defined by the presence of immune dysfunction. In brief, autoimmune disorders occur when the immune system misrecognizes host tissue as pathogenic and attempts to remove the misidentified host tissue [11]. In doing so, both a local and systemic inflammatory response is initiated. Locally, the immune system attempts to break down and clear the triggering tissue (e.g., local break down of skin in psoriasis). While triggering this local inflammatory response, pro-inflammatory cytokines are released and circulated systemically with some degree of penetration to the central nervous system (CNS) as well. As a group, autoimmune disorders have been identified to occur at increased rates in BD [6]. Epidemiological studies have consistently shown increased rates of inflammatory bowel disease (IBD), systemic lupus erythematosus (SLE), autoimmune thyroiditis, psoriasis, Guillain-Barré syndrome (GBS), autoimmune hepatitis, MS and rheumatoid arthritis (RA) in BD [12,13,14,15,16,17,18].

### 2.2. Bipolar Disorder and Chronic Infections

Infections are also classically associated with both a local and systemic inflammatory response. The inflammatory response to infections is an essential physiological response that has been evolutionarily conserved amongst all mammal species [11]; however, in the case of chronic infections, the prolonged inflammatory response may also have deleterious effects, as the immune response is best suited for clearing an acute infection [19]. Similar to autoimmune disorders, chronic infections may lead to chronic elevation of pro-inflammatory cytokines systemically and centrally. As such, an association between chronic infections and BD may be expected.

Dating back to the 19th century, there has been significant interest in the interaction between BD and chronic infections, such as Toxoplasma gondii (*T. gondii*), herpes simplex viru 1 (HSV1), cytomegalovirus (CMV) and human herpes virus 6 (HHV6); however, results have been mixed with poor replicability of identified associations [20]. The strongest replicated evidence has shown an increased co-prevalence of *T. gondii* in BD compared to the general population (odds ratio (OR) 1.52, *p* = 0.02) [21]. Interestingly, chronic infections, such as *T. gondii*, CMV and HSV1 have been associated with poorer cognitive function in BD [22,23]. Taken together, the association between BD and chronic infections remains unclear; however, BD patients with comorbid chronic infections may be at risk for a more severe phenotype secondarily to the presence of chronic low grade inflammation.

### 2.3. Bipolar Disorder and Cardiovascular Disease

Immune dysfunction is a key feature of cardiovascular disease (CVD) as inflammation plays a significant role in the progression of atherosclerotic plaques [24]. Cardiovascular disease has been strongly associated with BD in a bidirectional fashion. To emphasize the importance of this association, the American Heart Association (AHA) has recently recognized BD as an independent risk factor of early CVD. Indeed, replicated epidemiological studies have identified BD as an independent risk factor for CVD and vice versa [25,26,27,28,29,30]. Both cardiovascular and psychiatric researches have pointed to immune dysfunction as a likely key factor mediating this observed interaction. The high rate of comorbid BD and CVD is of particular importance because of its role in early mortality in BD; the increased prevalence of CVD is primarily responsible for the 10- to 20-year decrease in life expectancy in BD compared to the general population [31]. With this interaction in mind, some investigators have suggested that targeting immune dysfunction in this patient population may serve to simultaneously improve outcomes for BD, CVD and overall life expectancy [1,28].

### 2.4. Bipolar Disorder and Metabolic Disorders

Similar to CVD, immune dysfunction plays a key role in the progression of metabolic disorders [26,32,33]. Diabetes and central obesity have both been associated with chronic low grade inflammation, with the degree of inflammation being directly correlated with disease progression [34]. With immune dysfunction as a likely key mediating factor, BD has been strongly associated with increased rates of diabetes, obesity, dyslipidemia and metabolic syndrome [12,35,36].

A key factor facilitating chronic inflammation related to metabolic disorders is the presence of visceral adipose tissue (i.e., central obesity). Visceral adipose tissue is a direct source of chronic low-grade inflammation, increasing the production of pro-inflammatory adipokines and cytokines including IL-6, TNF-α, and C-reactive protein (CRP) [37,38]. Subcutaneous adipose tissue serves as a “metabolic sink” to prevent accumulation of visceral adipose tissue; however, under certain genetic (e.g., polygenic risk factors for central obesity) and environmental (e.g., sedentary lifestyle and poor diet) conditions, high volumes of dysfunctional visceral adipose tissue may accumulate [37,38]. In the context of chronic positive energy balance (e.g., greater caloric intake then expenditure), adipocytes undergo hypertrophy and have increased triglyceride stores [39]. The lypolytic rate is therefore increased leading to increased production of leptin (pro-inflammatory) and decreased production of adiponectin (anti-inflammatory), thereby signaling the release of pro-inflammatory cytokines [40]. Further, adipocyte hypertrophy promotes macrophage infiltration of adipose tissue. The resultant cross talk between macrophages and adipocytes promotes further release of pro-inflammatory cytokines and adipokines [37,38,39,40].

Bipolar disorder has also been associated with a slightly increased risk of developing gout [41]. With this epidemiological observation in mind, several investigators have recently hypothesized that purinergic system abnormalities and related variations of uric acid may be involved in the pathophysiology of BD [42,43]. Uric acid has been strongly associated with other metabolic disorders, increased oxidative stress and inflammation [44,45]. Further, several proof-of-concept clinical trials have identified a potential anti-manic effect of drugs lowering uric acid (e.g., allopurinol) [46].

## 3. Cytokine Changes Associated with Bipolar Disorder

Cytokines are signaling molecules of the immune system which may increase or decrease local and systemic inflammatory responses. Measuring cytokine levels peripherally (i.e., serum levels) and centrally (i.e., cerebral spinal fluid (CSF) levels) provides insight into immune system activity. Cytokine levels can identify current levels of inflammation and identify which specific part of the immune system is over or underactive leading to the observed immune dysfunction in BD. Moreover, as signaling molecules, specific cytokines may be directly implicated in the pathophysiology of BD and may therefore present as potential novel targets of treatment.

Cytokine levels have significant fluctuations and variability; however, some trends have emerged through numerous cytokine studies of BD patients compared to healthy controls [3,4]. These cytokine studies have consistently shown elevated levels of pro-inflammatory cytokines in BD, suggestive of chronic low grade inflammation. Serum levels of pro-inflammatory molecules including interleukin-4 (IL-4), tumor necrosis factor alpha (TNF-α), soluble interleukin-2 receptor (sIL-2R), interleukin-1 beta (IL-1β), interleukin-6 (IL-6), soluble receptor of TNF-α type 1 (STNFR1) and CRP are elevated in BD patients compared to healthy controls [3,47,48,49]. This cytokine profile indicates dysfunction of the *innate* immune system.

Another key observation has been variability in cytokine profiles depending on mood state (i.e., differing cytokine profiles during periods of depression, mania, hypomania and euthymia). This variability in cytokine profiles might suggest variable involvement of immune dysfunction in depression versus mania versus euthymia. Significant heterogeneity in BD cytokine studies has been problematic and, as such, there has been no clear cytokine profile that is reproducibly associated with each mood state [3,4]. This significant heterogeneity also suggests that inflammation is likely a pertinent pathogenic factor for only a *subset* of BD; this subset of BD may potentially represent an “inflammatory-BD” that may be pathophysiologically dissimilar from other BD patients. This potential sub-typing of BD is currently being investigated with important treatment implications.

Within the context of this substantial heterogeneity, the following mood-dependent cytokine profiles have been identified. The most robust evidence exists for an association between pro-inflammatory cytokines and depressive episodes, in both bipolar and unipolar depression [50]. During depressive episodes, serum levels of CRP, TNF-α, IL-6, IL-1β, sTNFR1 and CXCL10 are elevated [10,47,51]. Increased depression severity is associated with greater elevations of pro-inflammatory cytokines [52]. During manic episodes, serum levels of IL-6, TNF-α, sTNFR1, IL-RA, CXCL10, CXCL11 and IL-4 are often elevated [47,51]. During euthymic periods, sTNFR1 is the only consistently elevated inflammatory marker [47,48]. One significant limitation of these cytokine studies is their cross-sectional nature (i.e., serum levels are usually only taken at one point in time). Longitudinal studies are needed to measure cytokine levels within the same group of BD subjects to determine how they change during and in between mood episodes. Understanding this chronological relationship (e.g., if cytokines are elevated prior to versus after mood episode onset) would also provide further insight into the cross-talk between BD and immune dysfunction.

## 4. Pathophysiology of the Inflammatory-Mood Pathway

Numerous mechanisms have been identified which may mediate the bidirectional interaction between BD and immune dysfunction. Many of these mechanisms have been largely established in animal models [53]. More recently, clinical studies have provided evidence to suggest that these preclinical findings are valid in humans as well [10,54]. Herein we describe some of the key biological mechanisms which may contribute to the inflammatory-mood pathway. Of note, many of these mechanisms are not exclusive to BD and may trans-diagnostically sub-serve the interactions observed between immune dysfunction and other brain disorders (e.g., unipolar depression, schizophrenia, neurodegenerative disorders) [55]. Currently, it remains unclear the degree of overlap versus divergence in inflammatory processes mediating the interaction between immune dysfunction and various neuropsychiatric disorders [56]. We hypothesize that there are likely both trans-diagnostically shared immune pathways as well BD-specific immune pathways (i.e., immune changes and mechanism that may not be present in other disorders).

Central to the inflammatory-mood pathway is the ability of peripherally circulating cytokines to traverse the blood-brain-barrier (BBB). Systemically circulating cytokines may traverse the BBB via active transport channels and through leaky regions of the BBB [57]. Of note, the relative permeability of the BBB for various cytokines remains unclear; however, replicated evidence has demonstrated clear associations between elevation of cytokines in serum samples (i.e., peripherally circulating cytokines) with the same cytokines being elevated in cerebral spinal fluid (CSF) samples (i.e., cytokine levels in the CNS), suggesting that likely all cytokines may penetrate the CNS to some degree [47]. Recent findings in animal models have also suggested the presence of lymphatic vessels in the brain which could provide another direct pathway for cytokines and other signalling molecules to enter the CNS [58]. Cytokines may then signal several downstream effects which alter the structure and function of key brain regions sub-serving mood and cognitive function. Cytokines can directly alter monoamine levels, cause over-activation of microglial cells and lead to increased oxidative stress in the brain [53]. The net effect of these changes is neurodegeneration and decreased neuroplasticity in key brain regions which may lead to the phenotypic changes observed in BD and other brain disorders.

### 4.1. Cytokine-Induced Neurotransmitter Changes

Monoamine changes have been the focus of mood disorder research for many years. Further, the majority of psychiatric medications’ primary mechanism of action is through alteration of monoamine levels [59]. Pro-inflammatory cytokines may directly and indirectly alter monoamine levels in the CNS through numerous pathways. More specifically, TNF-α, IL-2 and IL-6 have been shown to directly alter monoamine levels [60]. IL-2 and interferon-gamma and alpha (IFN-γ and -α) increase the enzymatic activity of indolamine 2,3-dioxygenase (IDO), thereby increasing the breakdown of tryptophan to depressogenic tryptophan catabolites (TRYCATs). Serotonin (5-HT) levels may be further modulated through the IL-6 and TNF-α dependent breakdown of 5-HT to 5-hydroxyindoleacetic acid (5-HIAA) [61]. Depletion of tryptophan and decreased levels of 5-HT can directly impair affective and cognitive function [62].

Inflammation may also directly alter levels of dopamine and norepinephrine. Pro-inflammatory molecules, such as IFN, induce the activation of the guanosine-triphosphate-cyclohydrolase-1 (GTP-CH1) enzyme. Increased expression of GTP-CH1 results in the formation of neopterin and tetrahydrobiopterin (BH4), a cofactor used by phenylalanine hydroxylase (PH), tyrosine hydroxylase (TH) and tryptophan hydroxylase (TPH) to form tyrosine (Tyr), dopamine, norephinephrine, and serotonin, respectively; however, inflammation lowers pyruvoyl tetrahydropterin synthase (PTPS) activity, thus favouring neopterin formation instead of BH4 [63,64,65]. With decreased BH4 levels, the activity of PH, TH and TPH is decreased thus lowering the production of dopamine, norepinephrine and serotonin [55,66,67].

Taken together, pro-inflammatory signaling may decrease the levels of dopamine, norepinephrine and serotonin, which has long been associated with worsening mood and cognitive symptoms. Current pharmacotherapies target the end result of this pathway, namely, monoamine levels [59]. Targeting inflammation may have more disease modifying potential as immune dysfunction is “upstream” of the monoamine changes observed in mood disorders; correcting the underlying cause (i.e., immune dysfunction) may provide greater benefits than only treating symptomatically by correcting the downstream effect (i.e., monoamine changes).

Of recent interest has also been the potential interaction between inflammation and another key neurotransmitter, namely, glutamate. The importance of the glutamate system in mood disorder pathophysiology has been highlighted by the robust evidence demonstrating the rapid and potent antidepressant effects of ketamine, an N-methyl-d-aspartate (NMDA) glutamate receptor antagonist [68,69]. Significant cross-talk between glutamate and the immune system has now been demonstrated in pre-clinical and clinical models [70]. Inflammatory cytokines have been shown to influence glutamate metabolism through direct effects on microglia and astrocytes. As such, inflammatory cytokines may increase glutamate levels thus causing abnormal over-activation of glutamate receptors leading to uncontrolled increases of calcium influx through NMDA receptor channels, with the final result of excitotoxicity and impaired neuroplasticity [71].

The administration of exogenous pro-inflammatory cytokines has been shown to increase glutamate levels in the basal ganglia and anterior cingulate cortex (key brain regions sub-serving mood disorder pathology) as measured by magnetic resonance spectroscopy (MRS) [72]. Further, MRS studies in patients with unipolar depression have revealed that increased markers of inflammation (e.g., CRP) correlate with increased glutamate levels in the basal ganglia, which was specifically associated with anhedonia and psychomotor retardation [73]. In addition, an antidepressant response to ketamine may be predicted by elevated baseline inflammatory markers [74,75], further suggestive of significant cross-talk between immune dysfunction, the glutamate system and mood disorder pathophysiology.

### 4.2. Pathological Microglial Over-Activation

Microglia are the macrophages of the CNS that serve an important role in facilitating neuroplasticity [76,77,78]. Microglia aid in the pruning of unused neural circuits to allow for more space and energy to be made available for more frequently used neural circuits. Under physiological conditions, microglia may effectively prioritize the most important neural circuits leading to optimal brain structure and function [77,78]. However, with chronic inflammation, pro-inflammatory cytokines promote prolonged over-activation of microglia [76]. With this over-activation, microglia may aberrantly prune important neural circuits sub-serving mood and cognitive function (e.g., prefrontal cortex (PFC), amygdala, hippocampus, insula and the anterior cingulate cortex (ACC)) [76,79]. This process results in a positive feed-forward loop whereby activated microglia release cytokines, which further increases inflammation and further microglia recruitment and activation. The release of cytokines from activated microglia may also further perpetuate the previously discussed monoamine changes. Lastly, the over-activation of microglia increases the production of reactive oxygen species (ROS) leading to local oxidative stress, further damaging neural circuitry in key brain regions sub-serving mood and cognition [80]. This unfortunate cascade may contribute to the neuroprogression of BD as increasing numbers of important neural circuits are destroyed [47,81,82,83].

### 4.3. Inflammation and Increased Oxidative Stress

Oxidative stress has also been associated with mood disorders and is intimately connected with immune dysregulation, as inflammation increases oxidative stress and oxidative stress increases inflammation [84,85,86]. Oxidative stress occurs when there is an imbalance between the production of ROS and production of antioxidants responsible for neutralizing ROS [87]. Replicated evidence has demonstrated increased ROS and decreased antioxidants in BD, leading to pathologic neurodegeneration in key brain regions sub-serving mood and cognition [88,89,90]. Mood disorders have been associated with increased levels of pro-oxidant markers, namely, 8-hydroxy-2′-deoxyguanosine (8-OHdG), F2-isoprostanes, malondialdehyde (MDA) and decreased levels of anti-oxidant molecules, namely, glutathione (gamma-glutamyl-cysteinyl-glycine; GSH), superoxide dismutase (SOD) and glutathione peroxidase (GPx) [91]. Further, in unipolar depression, antidepressant response (to conventional antidepressants) has been associated with decreased oxidative stress, suggesting a mediational role of oxidative stress reduction in the effective treatment of mood disorders [87]. As such, there has been great interest in further understanding the mechanisms sub-serving increased oxidative stress along with the potential novel drug targets these mechanisms may offer.

### 4.4. Hypothalamic-Pituitary-Adrenal (HPA) Axis Over-Activation

Pro-inflammatory cytokines, namely IFN, TNF-α and IL-6, significantly up-regulate HPA activity thereby increasing systemic cortisol levels [92]. Under physiological conditions, HPA activation is advantageous to aid in the stress response required with an acute infection or injury. However, with chronic inflammation, HPA activation may be prolonged with deleterious effects related to chronic hypercortisolemia [93]. Additionally, chronic hypercortisolemia leads to downregulation of glucocorticoid receptor synthesis, translocation and sensitivity in the pituitary and hypothalamus, effectively inhibiting the negative feedback loop of the HPA axis [94]. This loss of the negative feedback loop leads to further propagation of hypercortisolemia with the well-established negative downstream effects (e.g., mood, cognitive and physical squealy) of chronically elevated cortisol levels [95,96,97,98]. Further, impaired cortisol suppression itself has long been recognized a strong predictor of mood disorders [98].

Dysfunction of the HPA axis has been identified in numerous medical and psychiatric disorders, however, the particular relevance in BD, in specific, was further emphasized by a recent meta-analysis and systematic review [99]. Belvederi Murri et al., (2016) identified forty-one studies showing that BD was consistently associated with significantly increased levels of cortisol (basal and post-dexamethasone) and adrenocorticotropic hormone (ACTH), but not of corticotropin-releasing hormone (CRH). These authors suggested that progressive HPA axis dysfunction is a putative mechanism that might underlie the clinical and cognitive deterioration of patients with BD and that targeting the HPA axis might be a novel strategy to improve the outcomes of BD [99].

### 4.5. The Microbiota-Gut-Brain Axis

In recent years, the role of the microbiota-gut-brain axis in neuropsychiatric disorders has become of great interest [100,101,102]. The gut and brain may communicate in a bidirectional fashion through numerous pathways including via the parasympathetic nervous system (primarily the vagus nerve), the gut neuroendocrine system, the circulatory system (delivering neuroactive metabolites and neuro-transmitters directly produced in the gut), and most notably, via the immune system [101]. The composition of the gut microbiota may have a large impact on the signaling molecules, including cytokines, that are being produced by the gastrointestinal (GI) system. The GI system may induce the production of pro-inflammatory cytokines on an acute or chronic basis. These cytokines may have direct effects on brain function as previously described.

Numerous investigators are questioning the potential impact of altering the gut microbiota on immune function and mental illness [103]. While this field is still in its infancy, the potential for novel treatments targeting the gut microbiota to treat BD may represent a completely new class of hypothesis-driven therapeutic interventions. For example, in a recent case report, Hamdani et al., (2015) suspected that a manic episode may have been triggered by alteration of the gut microbiota [104]. Given their hypothesis that the manic episode was triggered by perturbation of the gut-brain axis, the patient was treated with daily activated charcoal (a potent absorbent of gut inflammatory cytokines) instead of conventional anti-manic agents. The manic episode was successfully treated which corresponded to decreased serum levels of pro-inflammatory cytokines and chemokines. While targeting the microbiota to treat BD has yet to be assessed in any clinical trials, this case reports shows promise for a potential role of this novel target.

### 4.6. Inflammation and Sleep Dysfunction

Sleep dysfunction is a key feature of BD. During all phases of illness, changes in sleep patterns are commonly reported [105]. Indeed, during manic or hypomanic episodes, there is a characteristic decreased need for sleep. During depressive episodes, there may be difficulties achieving adequate quality or quantity of sleep or alternatively, hypersomnia in which patients are sleeping many more hours than would be typical for the general population. Even during euthymic periods, sleep complaints are still common in BD [105]. Sleep dysfunction is also strongly associated with immune dysfunction. Replicated evidence has demonstrated sleep dysfunction to be associated with increased levels of pro-inflammatory cytokines with a bidirectional causal association identified [106,107]. As such, interest has grown in immune dysfunction as a potential nexus sub-serving the bidirectional interaction between sleep dysfunction and BD [108,109].

## 5. Clinical Implications

Currently available treatments for BD have poor long term outcomes with high rates of treatment resistance and relapse [110]. Additionally, tolerability is often poor with significant adverse effects, such as weight gain and insulin resistance, being common with most evidence-based treatments [111]. Given the significant interaction between immune dysfunction and BD, the immune system presents as a potential novel target in the treatment of BD. The evidence discussed above suggests that inflammation may play a direct effect in the pathophysiology of BD in a subset of patients. Therefore, repurposing anti-inflammatory agents in the treatment of BD may potentially have disease modifying effects by targeting the underlying etiological processes rather than only treating symptomatically (i.e., the current approach).

Several proof-of-concept clinical trials have assessed the antidepressant effects of anti-inflammatory agents in the treatment of both unipolar [112] and bipolar [113] depression. In a recent meta-analysis conducted by our group to evaluate the antidepressant effects of anti-inflammatory agents, we identified eight randomized clinical trials (RCTs) (*n* = 312) assessing adjunctive nonsteroidal anti-inflammatory drugs (*n* = 53), omega-3 polyunsaturated fatty acids (*n* = 140), N-acetylcysteine, (*n* = 76), and pioglitazone (*n* = 44) in the treatment of BD. The overall effect size of adjunctive anti-inflammatory agents on depressive symptom severity was −0.40 (95% confidence interval −0.14 to −0.65, *p* = 0.002), indicative of a moderate antidepressant effect with good overall tolerability [113]. The clinical applicability of this meta-analysis was limited by the small number of studies included and small pooled sample size; however, this analysis provided further proof of concept that targeting the immune system may be an efficacious novel treatment for BD. Herein we further summarize clinical trials assessing specific anti-inflammatory agents in the treatment of BD.

### 5.1. N-Acetyl-Cysteine (NAC)

Among all anti-inflammatory agents, NAC has the strongest evidence as an adjunctive treatment for bipolar depression [114,115]. In an RCT of NAC for BD (*n* = 75), adjunctive NAC was shown to lower depression severity scores throughout the trial with a statistically and clinically significant difference compared to conventional therapy alone at the primary endpoint of 24 weeks [114]. Additionally, post-hoc analysis of 17 participants from this sample who met criteria for a current major depressive episode (MDE) at baseline revealed that 8 of 10 participants in the NAC group had a clinical response (i.e., greater than 50% reduction in depression severity) compared to only 1 of 7 participant in the placebo group [116]. An eight-week open-label trial of NAC also showed antidepressant effects in BD [117]. The effect of adjunctive NAC in mania/hypomania was also explored in a small post-hoc analysis of 15 BD participants experiencing an acute manic/hypomanic episode comparing participants receiving adjunctive NAC (*n* = 8) versus adjunctive placebo (*n* = 7). This analysis revealed a greater improvement in symptoms of mania in the NAC group compared to placebo [118]. Overall, NAC shows promise as an adjunctive treatment for BD during all phases of illness; however, evidence is strongest for use in the acute treatment of bipolar depression.

### 5.2. Omega-3 Polyunsaturated Fatty Acids (Omega-3s)

Several RCTs have also evaluated the effects of adjunctive omega-3s, a naturally-occurring and well-tolerated anti-inflammatory agent [119]. Results have been mixed with some trials showing an antidepressant effect in BD [120,121] and others reporting no antidepressant effect compared to conventional therapy alone [122,123,124]. When pooling these results together in a recent meta-analysis, a moderate and statistically significant anti-depressant effect of adjunctive omega-3s in BD was found compared to conventional therapy alone [119].

The mixed results of these studies assessing omega-3s in BD may suggest that omega-3s are beneficial in only a subset of BD. This hypothesis was further supported by a recent study assessing the antidepressant effects of omega-3s in the treatment of unipolar depression [125]. In this RCT, omega-3s were found to have a significant antidepressant effect in participants with elevated inflammatory markers. Intriguingly, in participants with normal cytokine levels, placebo had a greater antidepressant effect, compared to omega-3s, leading to an overall negative study outcome (i.e., no significant antidepressant effect was found when including the entire sample). While this study was in unipolar depression, it is likely that a similar effect may be observed in BD, in that only patients with elevated inflammatory markers may benefit from omega-3s, however, further study is still required to confirm or refute this hypothesis in the BD population.

### 5.3. Non-Steroidal Anti-Inflammatory Drugs (NSAIDs)

The anti-depressant effect of adjunctive NSAIDs has also been evaluated in BD. Nery et al., assessed adjunctive celecoxib in BD (*n* = 28) during an acute depressive or mixed episode [126]. Adjunctive celecoxib lowered depression severity by week 1; however, the primary outcome was negative as change in depression severity converged with the placebo group by the end of week 6. Saroukhani et al., assessed the effect of adjunctive aspirin in an RCT with male BD patients (*n* = 32) and found no significant difference between treatment groups by the primary endpoint of 6-weeks [127].

Three studies have also evaluated the effect of NSAIDs during acute manic/hypomanic episodes. In a small, proof-of-concept RCT, Arabzadeh et al., compared adjunctive celecoxib to treatment as usual for acute mania in BD inpatients (*n* = 46) [128]. They observed a significantly higher remission rate in the celecoxib group (87.0%) compared to the placebo group (43.5%) by the week 6 primary endpoint (*p* = 0.005). The same investigators also evaluated adjunctive celecoxib in an RCT of adolescent inpatients (*n* = 42) during an acute manic episode [129]. There was no significant difference in remission rates by the primary endpoint of 8-weeks, however, significantly greater improvement was observed in Young Mania Rating Scale (YMRS) scores in the celecoxib group compared with the placebo group by the week 8 primary endpoint (*p* = 0.04). In another RCT including BD inpatients (*n* = 35) with mania receiving electroconvulsive therapy (ECT), participants received either celecoxib or placebo from one day before the first ECT session throughout the sixth session. Brain-derived neurotrophic factor (BDNF) levels were also measured before and during the trial. Adding celecoxib was not associated with a significant rise in BDNF levels following ECT. No difference was noted between groups in terms of treatment response [130].

Taken together, the effect of NSAIDs in bipolar depression remains unclear as clinical studies have yielded mixed results. Additionally, adjunctive NSAIDs in the treatment of mania has yielded mixed results with anti-manic effects yet to be consistently demonstrated.

### 5.4. Minocycline

Minocycline is a tetracycline antibiotic with potent anti-inflammatory and neuroprotective effects [131]. Since the first case report of minocycline for bipolar depression was published in 1996 [132], there has been significant interest and off-label prescribing of minocycline for bipolar and unipolar depression; however, until this year (2017) there were no published RCTs to support or refute the antidepressant effects of minocycline. Recently, several open label trials and RCTs have been conducted to evaluate the antidepressant effects of minocycline for bipolar and unipolar depression [133,134,135,136,137]. In a recently published pilot, open-label, 8-week study, Soczynska et al., (2017) evaluated the efficacy, safety and tolerability of adjunctive minocycline for the treatment of bipolar I/II depression [134]. Adjunctive minocycline was associated with a significant reduction in depressive symptom severity from baseline to week 8 with overall good tolerability. While there has yet to be an RCT of minocycline for bipolar depression, these results show promise for a significant antidepressant effect and merit further investigation.

### 5.5. TNF-α Inhibitors

TNF-α inhibitors have also been of interest as they may directly target a key cytokine (i.e., TNF-α) known to be implicated in the inflammatory-mood pathway. One pivotal RCT assessed infliximab in treatment resistant depression (*n* = 60), including both bipolar and unipolar depressed patients in their sample. Although the overall antidepressant effect was negative for this study, a significant antidepressant effect was observed for a subgroup of participants, namely, those with elevated levels of serum CRP and TNF-α [138]. Similar to the previously discussed omega-3 RCT [125], the results of this trial suggested that stratification using inflammatory biomarkers might help determine which patients may benefit from anti-inflammatory treatments. A 12-week RCT evaluating the effects of adjunctive infliximab for the treatment of BD patients with elevated inflammatory markers is currently underway, directly implementing this type of stratified approach (NCT02363738).

### 5.6. Anti-Inflammatory Effects of Conventional Mood Stabilizers

Also of interest has been understanding the relative impact of conventional mood stabilizers on the immune system. Indeed, as previously discussed, the initial hypothesis of conceptualizing BD as an immune disorder was developed through observing the immune-modulating effects of lithium, one of the oldest and most effective treatments of BD [2]. The interaction between lithium and the immune system is complex as lithium has been shown to have both anti-inflammatory (e.g., suppression of cyclooxygenase-2 expression, inhibition of IL-1β and TNF-α production, and enhancement of IL-2 and IL-10 synthesis) and pro-inflammatory effects (e.g., induction of IL-4, IL-6 and other pro-inflammatory cytokines synthesis) [139,140]. As such, the ‘net effect’ of lithium on immune function may vary greatly; however, long term lithium use has been associated with normalization of cytokine levels [141].

Compared to lithium, much less in know about the impact of valproic acid on the immune system. Pre-clinical studies have suggested possible anti-inflammatory effects of valproic acid, however, clinical studies have failed to demonstrate a significant anti-inflammatory effect, as determined by changes in cytokine levels pre- and post-treatment [142,143]. The impact of carbamazepine, lamotrigine and antipsychotics on the immune system also remains unclear due to a lack of clinical studies [141].

## 6. Conclusions

Bipolar disorder is strongly associated with immune dysfunction. Moreover, in a subset of BD, immune dysfunction is likely playing a key role in the pathophysiology of disease progression. The bidirectional interaction of BD with immune dysfunction is likely responsible for the high rates of inflammatory comorbidities, such as autoimmune disorders, cardiovascular disease and metabolic disturbances. This interaction is of particular importance as medical comorbidity is primarily responsible for early mortality in BD. Numerous biological mechanisms of the inflammatory-mood pathway have been identified that may present novel targets in the treatment of BD. Targeting the immune system shows promise for improving BD outcomes as it may allow for disease modification through treatment of the underlying etiology (i.e., immune dysfunction), rather than only superficially treating the downstream effects as symptoms arise. Numerous proof-of-concept clinical trials have demonstrated a positive effect of anti-inflammatory agents in BD with generally good tolerability. Currently available evidence suggests that anti-inflammatory agents may be specifically helpful in the treatment of bipolar depression. Conversely, the impact of anti-inflammatory agents in mania and hypomania remains unclear. Clinical studies have also suggested that anti-inflammatory agents may be only beneficial for a subset of BD patients, namely, patients with immune dysfunction, as indicated by elevation of inflammatory markers. As such, future clinical trials should stratify patients based on inflammatory profile to determine which specific anti-inflammatory agent(s) are efficacious in which specific subset of BD patients.

## Figures and Tables

**Figure 1 brainsci-07-00144-f001:**
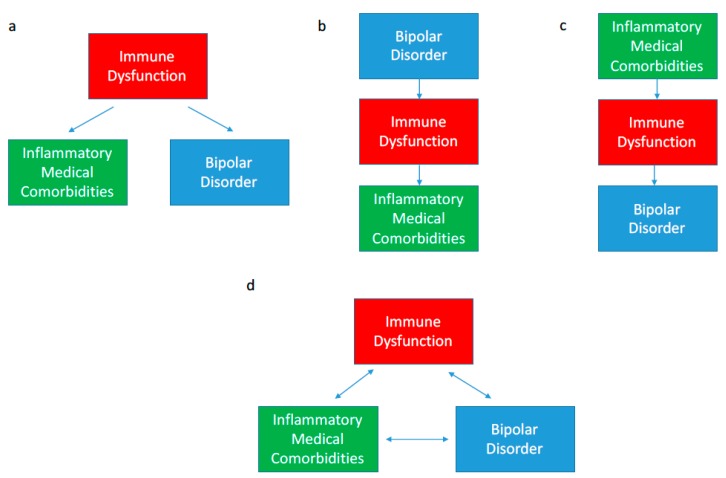
Potential interactions between bipolar disorder (BD), immune dysfunction and inflammatory comorbidities. (**a**) Immune dysfunction may be a common underlying cause of both BD and an inflammatory comorbidity; (**b**) BD may proceed the inflammatory condition or (**c**) vice versa. All three scenarios are observed in the BD population suggesting that the interaction is likely bidirectional in that immune dysfunction, BD and inflammatory comorbidities may be perpetuating each other (**d**).

**Table 1 brainsci-07-00144-t001:** Inflammatory comorbidities associated with bipolar disorder, as shown by epidemiological studies.

Category	Specific Conditions
**Autoimmune disorders**	Inflammatory bowel disease (IBD)
	Systemic lupus erythematosus (SLE)
	Autoimmune thyroiditis
	Guillain-Barré syndrome (GBS)
	Autoimmune hepatitis
	Rheumatoid arthritis (RA)
	Multiple sclerosis (MS)
	Psoriasis
**Chronic infections**	Toxoplasma gondii (*T. gondii*),
	Possibly herpes simplex virus 1 (HSV1),
	cytomegalovirus (CMV) and human herpes virus 6 (HHV6)
**Cardiovascular disease**	Myocardial infarction
	Stroke
	Atherosclerosis
	Hypertension
**Metabolic disorders**	Type II diabetes mellitus
	Dyslipidemia
	Central obesity
	Metabolic syndrome
	Gout

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
