# Peer review of "Bipolar Disorder and Immune Dysfunction: Epidemiological Findings, Proposed Pathophysiology and Clinical Implications"

_brainsci, 2017, doi:10.3390/brainsci7110144_

Round 1

Reviewer 1 Report

This is, in summary, an interesting manuscript aimed to review the current literature about the link between bipolar disorder (BD) and immune dysfunctions. According to existing cytokine studies, the authors reported that BD is associated with chronic low-grade inflammation with further increases in pro-inflammatory cytokine levels during mood episodes. The authors highlighted as key mechanisms underlying this link: the cytokine-induced monoamine changes, increased oxidative stress, pathological microglial over-activation, hypothalamic-pituitary-adrenal axis over-activation (HPA), alterations of the microbiome-gut-brain axis as well as sleep-related immune changes. They concluded that immune dysfunctions are likely to play a role only in a small subset of BD patients providing important insights into the pathophysiological mechanisms of this complex condition.

The authors may find as follows my main comments/suggestions.

First, when throughout the “Bipolar Disorder and Inflammatory Comorbidities” section, the authors reported that when an inflammatory comorbidity is present, peripherally released pro-inflammatory cytokines may increase systemic cytokine levels throughout the body including in the brain”, specifically to which type of pro-inflammatory cytokines the authors referred to? Here, more details/information may be useful for the general readership. This is particularly relevant as the same authors more ahead stated that while triggering the local inflammatory response inautoimmune disorders, pro-inflammatory cytokines are released and circulated systemically with some degree of penetration to the central nervous system.

In addition, i suggest to update the section regarding the supposed link between BD and increased rates of diabetes, obesity, dyslipidemia and metabolic syndrome as only rapidly hypothesized by the authors. Here, the most relevant mechanisms underlying this association should be provided in a more detailed manner. In particular, how inflammation may be directly correlated with disease progression could be more deeply discussed.

Importantly, when the authors correctly stated that cytokines may directly alter monoamine levels causing an over-activation of microglial cells and leading to increased oxidative stress in the brain, they could add that the abnormal activation of glutamate receptors leading to the uncontrolled Ca2+ influx through NMDA receptor channels with the final result of excitotoxicity and impaired neuroplasticity may be enhanced by the abnormally elevated concentrations of inflammatory cytokines together with neuroinflammation. It has been recently reported that some glutamate antagonists such as ketamine may act neutralizing these abnormally elevated inflammatory cytokines levels. Ketamine also relieves depressive symptoms within hours from administration and it is also handful in managing patients resistant to treatment with other drugs. Most studies demonstrated the role of NMDA antagonists such as ketamine in reducing neuroinflammation and abnormally increased inflmmatory cytokines levels, confirming the active role of glutamate in the pathophysiology of this complex condition. In order to address this relevant issue, i suggest to cite and discuss the paper of Serafini and colleagues which has been published on Current Neuropsychopharmacology in 2014.

Furthermore, some statements such as “Oxidative stress occurs when there is an imbalance between the production of ROS and production of antioxidants responsible for neutralizing ROS” are interesting as reported but need to be supported by adequate references. Moreover, when the authors reported that antidepressant response has been associated with decreased oxidative stress, to which type of antidepressant medications the authors exactly referred?

Moreover, as within the “Hypothalamic-Pituitary-Adrenal (HPA) Axis Over-Activation” section the authors correctly mentioned that with chronic inflammation, HPA activation may be prolonged with deleterious effects related to chronic hypercortisolemia, they could further stress the relevance of HPA dysfunctions in bipolar disorder. Recent evidence documented that HPA axis dysfunctions are closely involved in the pathophysiology of many diseases, in particular bipolar disorder with important pathophysiological implications. Targeting HPA axis dysfunctions might be a novel strategy to improve the outcomes of these conditions. The recent systematic review/metanalysis of Belvederi Murri and colleagues which was published on Psychoneuroendocrinology in 2016 may be cited and discussed.

Finally, the main limitations/shortcomings of the present review should be described as they are lacking according to the current version of the paper.  

Author Response

Thank you for your thorough review of our submission.  We appreciate your feedback and comments.  We believe your comments serve to improve the overall quality of the submission and have implemented all suggestions as follows:

-we have added examples of specific cytokines released peripherally that increase systemic levels and penetrate the CNS in section 2.  We have also added further information about the relative penetration of the CNS of various cytokines into section 4

-we have added more details about the interaction between metabolic disorders and chronic inflammation as suggested

-we have added a discussion of the interplay between glutamate and the immune system to section 4.1 and changed the title of this section to ‘Cytokine-Induced Neurotransmitter Changes’ to allow for discussion of monoamines as well as glutamate

-further details and references regarding oxidative stress have been added as suggested

-as suggested, we have added further discussion of the HPA axis and the recent meta-analysis by Belvederi Murri

-At the end of the introduction, we have commented on the limitations and shortcomings of the present review.

Reviewer 2 Report

I read with great interest the valuable and authoritative article by Rosenblat and McIntyre, overviewing immune dysfunctions in bipolar disorder. The authors summarized main data on the potential association between bipolar disorder and immune system, summarizing relevant epidemiological data and findings from studies exploring cytokine levels, and discussing potential pathophysiological mechanisms and treatment implications. This topic is an important area of research updating previous important reviews carried out by the same authors [1,2]. Despite the manuscript can be accepted also in its current form, the authors can complement it according to the following minor comments:

* The authors rightly reported that immune dysfunctions are probably «not exclusive to bipolar disorder and may transdiagnostically sub-serve the interactions observed between immune dysfunction and other brain disorders (e.g. unipolar depression, schizophrenia, neurodegenerative disorders)». More information may be added about this concept, clarifying if there might be a ‘specificity’ of immune dysfunctions in bipolar disorder or if it should be considered completely or partially overlapped with those occurring in other psychiatric disorders, such as schizophrenia. This issue is important, since despite bipolar disorder and schizophrenia may share a similar immune profile, higher rates of organ-specific autoantibodies have been reported in bipolar disorder, possibly supporting the hypothesis of its autoimmune pathogenesis [3].

It has been recently hypothesized that purinergic system abnormalities [4] and related variations of uric acid [5] may be involved in the pathophysiology of bipolar disorder and related phases. Indeed, it has been shown a potential therapeutic effect of drugs lowering uric acid (allopurinol) for treating mania [6]. In addition, bipolar disorder has been associated with slightly higher risk of gout [7]. This may be an important issue, since uric acid has been associated with metabolic disorders (e.g., [8]), oxidative stress (e.g., [9]) and inflammatory state (e.g., [10]). I guess that these concepts can be added in the 2.4 paragraph (‘Bipolar Disorder and Metabolic Disorders’).

* The authors provided a useful overview of anti-inflammatory agents that may be potentially efficacious in treating bipolar disorder. However, it would be useful if the authors could add more information on potential effects on immune system of standard treatments for bipolar disorder, such as mood stabilizers (e.g., [11,12]).

References

[1]         Rosenblat JD, McIntyre RS. Are medical comorbid conditions of bipolar disorder due to immune dysfunction? Acta Psychiatr Scand. 2015; 132: 180-191.

[2]         Rosenblat JD, McIntyre RS. Bipolar Disorder and Inflammation. Psychiatr Clin North Am. 2016; 39: 125-137.

[3]         Davison K. Autoimmunity in psychiatry. Br J Psychiatry. 2012; 200: 353-355.

[4]         Cheffer A, Castillo ARG, Corrêa-Velloso J, Gonçalves MCB, Naaldijk Y, Nascimento IC, Burnstock G, Ulrich H. Purinergic system in psychiatric diseases. Mol Psychiatry. 2017. doi: 10.1038/mp.2017.188.

[5]         Bartoli F, Crocamo C, Mazza MG, Clerici M, Carrà G. Uric acid levels in subjects with bipolar disorder: A comparative meta-analysis. J Psychiatr Res. 2016; 81: 133-139.

[6]         Bartoli F, Crocamo C, Clerici M, Carrà G. Allopurinol as add-on treatment for mania symptoms in bipolar disorder: systematic review and meta-analysis of randomised controlled trials. Br J Psychiatry. 2017; 210: 10-15.

[7]         Chung KH, Huang CC, Lin HC. Increased risk of gout among patients with bipolar disorder: a nationwide population-based study. Psychiatry Res. 2010; 180: 147-150.

[8]         Yuan H, Yu C, Li X, Sun L, Zhu X, Zhao C, Zhang Z, Yang Z. Serum Uric Acid Levels and Risk of Metabolic Syndrome: A Dose-Response Meta-Analysis of Prospective Studies. J Clin Endocrinol Metab. 2015; 100: 4198-4207.

[9]         Glantzounis GK, Tsimoyiannis EC, Kappas AM, Galaris DA. Uric acid and oxidative stress. Curr Pharm Des. 2005; 11: 4145-4151.

[10]      Martinon F. Update on biology: uric acid and the activation of immune and inflammatory cells. Curr Rheumatol Rep. 2010; 12: 135-141.

[11]      Maddu N, Raghavendra PB. Review of lithium effects on immune cells. Immunopharmacol Immunotoxicol. 2015; 37: 111-125.

[12]      van den Ameele S, van Diermen L, Staels W, Coppens V, Dumont G, Sabbe B, Morrens M. The effect of mood-stabilizing drugs on cytokine levels in bipolar disorder: A systematic review. J Affect Disord. 2016; 203: 364-373.

Author Response

Thank you for your thorough review of our submission.  We appreciate your feedback and comments.  We believe your comments serve to improve the overall quality of the submission and have implemented all suggestions as follows:

-we have further discussed shared versus divergent mechanisms involved in the interaction between immune dysfunction and various psychiatric disorders

-we have added a discussion of gout and uric acid into section 2.4

-we have added a section (5.6 Anti-inflammatory Effects of Conventional Mood Stabilizers) on the impact of mood stabilizers on the immune system